# Parametrizing Product Shape Manifolds by Composite Networks

**Josua Sassen**
University of Bonn

**Klaus Hildebrandt**
TU Delft

**Martin Rumpf**
University of Bonn

**Benedikt Wirth**
University of Münster

## Abstract

Parametrizations of data manifolds in shape spaces can be computed using the rich toolbox of Riemannian geometry. This, however, often comes with high computational costs, which raises the question if one can learn an efficient neural network approximation. We show that this is indeed possible for shape spaces with a special product structure, namely those smoothly approximable by a direct sum of low-dimensional manifolds. Our proposed architecture leverages this structure by separately learning approximations for the low-dimensional factors and a subsequent combination. After developing the approach as a general framework, we apply it to a shape space of triangular surfaces. Here, typical examples of data manifolds are given through datasets of articulated models and can be factorized, for example, by a Sparse Principal Geodesic Analysis (SPGA). We demonstrate the effectiveness of our proposed approach with experiments on synthetic data as well as manifolds extracted from data via SPGA.

## 1 Introduction

Modeling collections of shapes as data on Riemannian manifolds has enabled the usage of a rich set of mathematical tools in areas such as computer graphics and vision, medical imaging, computational biology, and computational anatomy. For example, Principal Geodesic Analysis, a generalization of Principal Component Analysis, can be used to parametrize submanifolds approximating given data points while preserving structure of the data such as its invariance to rigid motion. The evaluation of such a parametrization, however, typically comes at a high computational cost as the Riemannian exponential, mapping infinitesimal shape variations to shapes, has to be evaluated. This motivates trying to learn an efficient approximation for these parametrizations. Direct application of deep neural networks (NNs), however, proves ineffective for high-dimensional spaces with strongly nonlinear variations. Therefore, we consider more structured shape manifolds, namely, we assume that they can be approximated by an affine sum of low-dimensional submanifolds. In computer graphics, typical examples of data manifolds are given through datasets of articulated models, *e.g.* human bodies, faces or hands. Then, the desired structure of an affine sum of factor manifolds can be produced, for example, by a Sparse Principal Geodesic Analysis (SPGA). Motivated by this, we exploit the data manifolds' approximability with such affine sums: We separately approximate the exponential map on the factor manifolds by fully connected NNs and the subsequent combination of factors by a convolutional NN to yield our approximate parametrization. In formulas, based on a judiciously chosen decomposition $v = v_1 + \ldots + v_J$, our aim is to approximate the Riemannian exponential $\exp_z(v)$ by $\Psi^\zeta(\psi_1^\zeta(v_1), \ldots, \psi_J^\zeta(v_J))$, where $\Psi^\zeta$ is a NN and the $\psi_j^\zeta$ are further NNs approximating the Riemannian exponential $\exp_z$ on the low-dimensional factor manifolds.

We develop our approach focusing on the shape space of discrete shells, where shapes are given by triangle meshes and the manifold is equipped with an elasticity-based metric. In principle, our approach is also applicable to other shape spaces such as manifolds of images, and we will include remarks on how we propose this could work. We evaluate our approach with experiments on data manifolds of triangle meshes, both synthetic ones and ones extracted from data via SPGA, and we demonstrate that the proposed composite network architecture outperforms a monolithic fully connected network architecture as well as an approach based on the affine combination of the factors. We see this work as a first step to use NNs to accelerate the complex computations of shape manifold parameterizations. Therefore, we think that our approach has great potential to stimulate further research in this direction, which could in turn advance the applications of Riemannian shape spaces.

**Contributions**    In summary, the contributions of this paper are

- combining the Riemannian exponential map on shape spaces and neural network methodology for the efficient parametrization of shape space data manifolds,
- demonstrating the applicability of such an approach for data manifolds which can be smoothly approximated via direct sums of low-dimensional submanifolds,
- using a combination of fully connected neural networks for the factorwise Riemannian exponential maps and a convolutional network to couple them,
- verifying that such a setup works well with existing methods to construct product manifolds, such as Sparse Principal Geodesic Analysis, and
- showing that the composite network architecture outperforms alternative approaches.

## 2    RELATED WORK

**Shape Spaces**    Shape spaces are manifolds in which each point is a shape, *e.g.*, a triangle mesh or an image. A Riemannian metric on such a space provides means to define distances between shapes, to interpolate between shapes by computing shortest geodesic paths, and to explore the space by constructing the geodesic curve starting from a point into a given direction. Shape spaces have proven useful for applications in areas such as computer graphics (Kilian et al., 2007; Heeren et al., 2012; Wang et al., 2018) and vision (Heeren et al., 2018; Xie et al., 2014), medical imaging (Kurtek et al., 2011b; Samir et al., 2014; Kurtek et al., 2016; Bharath et al., 2018), computational biology (Laga et al., 2014), and computational anatomy (Miller et al., 2006; Pennec, 2009; Kurtek et al., 2011a). For an introduction to the topic, we refer to the textbook of Younes (2010).

**Shape Space of Meshes**    Triangle meshes are widely used to represent shapes in computer graphics and vision. Riemannian metrics on shape spaces of triangle meshes can be defined geometrically, using norms on function spaces on the meshes (Kilian et al., 2007), or physics-based, considering the meshes as thin shells and measuring the dissipation required to deform the shells (Heeren et al., 2012; 2014). The computation of geodesic curves in these spaces requires numerically solving high-dimensional nonlinear variational value problems, which can be costly. For shape interpolation problems, model reduction methods can be used to efficiently find approximate solutions (Brandt et al., 2016; von Radziewsky et al., 2016).

**Statistics in Shape Spaces**    Data in a Riemannian shape spaces can be analyzed using Principal Geodesic Analysis (PGA) (Fletcher et al., 2004; Pennec, 2006). Analogous to principal component analysis (PCA) for data in Euclidean spaces, PGA can construct low-dimensional latent representations that preserve much of the variability in the data. This is achieved by mapping the data with a non-linear mapping, the Riemannian logarithmic map, from the manifold to a linear space, the tangential space at the mean shape, and computing a PCA there. Latent variables of the PCA are then mapped with the inverse mapping, the Riemannian exponential map, onto the manifold, so that the latent space describes a submanifold of the shape space. A PGA in shape spaces of meshes was introduced in (Heeren et al., 2018) and used to obtain a low-dimensional, nonlinear, rigid body motion invariant description of shape variation from data.

**Sparse PGA**    While PCA modes involve all variables of the data, Sparse Principal Component Analysis (Zou et al., 2006) constructs modes that involve just few variables. This is achieved by integrating a sparsity encouraging term to the objective that defines the modes. Based on this idea, Neumann et al. (2013) proposed a scheme for extracting Sparse Localized Deformation Components (SPLOCS) from a dataset of non-rigid shapes. Since SPLOCS are linear modes, they are well-suited to describe small deformations such as face motions accurately. To increase the range of deformations and to compensate linearization artifacts Huang et al. (2014) integrated SPLOCS with gradient-domain techniques and Wang et al. (2017; 2021) with edge lengths and dihedral angles. In (Sassen et al., 2020b), a Sparse Principal Geodesic Analysis (SPGA) was introduced. Similar to PGA, the SPGA modes are nonlinear and rigid motion invariant. On top of that, however, the SPGA modes describe localized deformations. Due to the localization, many pairs of SPGA modes have disjoint support and are therefore independent of each other. We want to take advantage of this property to effectively learn the reconstruction of points in the manifold from their latent representation through an adapted network structure.

**Product Manifolds**   This work is focused on the parametrization of the product manifold structures obtained from SPGA. Alternative approaches for extracting product structures of data manifolds include the approach of Fumero et al. (2021), which finds a product structure of a manifold based on a geometric notion of disentanglement, the Geometric Manifold Component Estimator presented in (Pfau et al., 2020), which uses a Lie group of transformations to generate a symmetry-based disentanglement of data manifolds, and the approach of Zhang et al. (2021), which decomposes the eigenfunctions of the Laplace–Beltrami operator of a manifold in order to find a product structure in the manifold. To the best of our knowledge, there are no works focusing on using networks to approximate the parametrization of such product manifolds in Riemannian shape spaces.

## 3   PRELIMINARIES AND NOTATION

We introduce step by step the background necessary to understand the context of our work. We also provide an overview of our notation in Appendix E.

**Riemannian Shape Space**   A shape space is a manifold $\mathcal{S} \subset \mathbb{R}^n$ whose elements are shapes. These could, for example, be images, curves, or surfaces described in various ways. Endowing such a shape manifold with the structure of a *Riemannian* manifold, *i.e.* a (smoothly) $z$-dependent inner product $g_z$ on the tangent space $T_z\mathcal{S}$ at each point $z \in \mathcal{S}$, provides us with a rich set of geometric tools. For example, we can use geodesics $c\colon [0,1] \to \mathcal{S}$, *i.e.* arc length parametrized locally shortest paths, as mathematical formulation of shape interpolation. The Riemannian logarithm $\log_z \tilde{z} \in T_z\mathcal{S}$ is then defined as the time derivative $\dot{c}(0)$ of the geodesic $c$ interpolating between $c(0) = z$ and $c(1) = \tilde{z}$. This allows to interpret the tangent space $T_z\mathcal{S}$ as the linear space of infinitesimal shape variations. Lastly, the Riemannian exponential map is the inverse of the logarithm, which means that for a tangent vector $v \in T_z\mathcal{S}$ one 'shoots' a geodesic in its direction, *i.e.* constructs a geodesic curve $c$ with initial velocity $v$ to obtain $\exp_z v \coloneqq c(1) \in \mathcal{S}$. The exponential map allows to transfer operations from infinitesimal shape variations back to actual shapes. We provide more details on Riemannian operators and their discretization in Appendix C.

**Principal Geodesic Analysis**   With these tools at hand, one can use Principal Geodesic Analysis (PGA) to compute submanifolds of the shape space approximating given data points $\{z^i\} \subset \mathcal{S}$: One first computes their Riemannian center of mass $\bar{z}$, *i.e.* the point with minimal sum of squared distances to all data points. Then one computes the logarithms $v^i = \log_{\bar{z}} z^i$ and thus linearizes the approximation problem at the center of mass by passing to the tangent space $T_{\bar{z}}\mathcal{S}$. In $T_{\bar{z}}\mathcal{S}$, one uses classical Principal Component Analysis (PCA) to compute the $m$ dominant modes $\{u^j\}$, whose span $\mathcal{U}$ is the best $m$-dimensional subspace approximating the logarithms $v^i$. Then the submanifold $\mathcal{M}$ approximating the data points is parametrized by the exponential map, *i.e.* $\mathcal{M} \coloneqq \exp_{\bar{z}} \mathcal{U}$.

**Sparse Principal Geodesic Analysis**   The coordinates on Riemannian shape spaces, *e.g.* pixel values, often correspond to different spatial locations of the shape, *e.g.* pixel positions, such that it makes sense to consider their support and sparsity: Sassen et al. (2020b) introduced Sparse Principal Geodesic Analysis (SPGA) to compute spatially localized dominant modes with widely disjoint supports. Those modes are easier to interpret semantically and in addition allow for efficient approximations of the exponential map. SPGA is performed by adding an appropriate sparsity-inducing regularization functional $\mathcal{R}$ to the variational formulation of PCA to compute such sparse deformation modes. The concrete choice of $\mathcal{R}$ depends on the specific shape space. Hence, for a given set $V \in \mathbb{R}^{n \times K}$ of $K$ logarithms, the SPGA problem to compute the first $m$ dominant modes $U \in \mathbb{R}^{n \times m}$ reads

$$\underset{\substack{U \in \mathbb{R}^{n \times m} \\ W \in \mathbb{R}^{m \times K}}}{\text{minimize}} \quad \|V - UW\|_g^2 + \lambda\,\mathcal{R}(U) \tag{1}$$

$$\text{subject to} \quad u_j \in T_{\bar{z}}\mathcal{S} \ \text{ and } \ |w_j|_\infty \leq 1 \text{ for } j \in \{1, \ldots, m\}.$$

The bound on the magnitude of the weights ensures that the coordinates of the $u_j$ do not shrink while the weights grow inverse proportionally. As before, the manifold approximating the data points is then parametrized using the exponential map. It can be equipped with a product structure by grouping the modes and applying the exponential map to the corresponding subspaces. This will be elucidated more in Section 4.

**NRIC Manifold**   As shapes we will consider triangular surfaces with fixed connectivity, *i.e.* with shared sets of vertices $\mathcal{V}$, edges $\mathcal{E}$, and faces $\mathcal{F}$. For vertex positions $X \in \mathbb{R}^{3|\mathcal{V}|}$, we denote by

$l(X) = (l_e(X))_{e \in \mathcal{E}}$ the vector of edge lengths and by $\theta(X) = (\theta_e(X))_{e \in \mathcal{E}}$ the vector of dihedral angles. We consider the vectors $z(X) = (l(X), \theta(X)) \in \mathbb{R}^{2|\mathcal{E}|}$ combining both. The manifold of all $z \in \mathbb{R}^{2|\mathcal{E}|}$ corresponding to immersed triangular surfaces is given by

$$\mathcal{S} := \{ z \in \mathbb{R}^{2|\mathcal{E}|} \, \big| \, \mathcal{T}(z) > 0, \, \mathcal{Q}(z) = 0 \}, \tag{2}$$

where $\mathcal{T}(z) > 0$ encodes the triangle inequalities and $\mathcal{Q}(z) = 0$ are the discrete integrability conditions from Wang et al. (2012), see also Sassen et al. (2020a) for more details. $\mathcal{S}$ is called the NRIC manifold, short for Nonlinear Rotation-Invariant Coordinates, and can be equipped with an elasticity-based Riemannian metric (Heeren et al., 2014). To evaluate the logarithm and exponential map, we use the time-discretization developed by Rumpf & Wirth (2015).

To obtain vertex positions for given lengths and angles, we use the nonlinear least-squares method from Fröhlich & Botsch (2011) based on $L^p$-norms to measure the difference between arbitrary $z$,

$$\|z^a - z^b\|_{p,\bar{z}}^p := \sum_{e \in \mathcal{E}} w_{e,l}^p |l_e^b - l_e^a|^p + \eta \sum_{e \in \mathcal{E}} w_{e,\theta}^p |\theta_e^b - \theta_e^a|^p. \tag{3}$$

The weights are computed from a reference shape $\bar{z}$ with vertex positions $\bar{X}$, *e.g.* the center of mass from above, as $w_{e,l} = l_e(\bar{X})^{-1}$ and $w_{e,\theta} = l_e(\bar{X}) a_e(\bar{X})^{-1/2}$, where $a_e$ is (one third of) the area of both faces adjacent to $e$. The bending weight $\eta$ is the same as used in the elasticity-based metric.

To apply SPGA to this shape space, we need to specify the sparsity-inducing regularization $\mathcal{R}$. Sassen et al. (2020b) observed that the simple mode-wise $L^1$-norms

$$\mathcal{R}(U) = \sum_{j=1}^m \|u_j\|_1 \tag{4}$$

suffice due to the natural connection of NRIC variation with elastic distortions. We provide more details on NRIC in Appendix D.

**Neural Networks**  We indicate functions that are implemented as neural networks by the superscript $\zeta$ as in $\varphi^\zeta$. This $\zeta$ represents the network parameters and is the same for all occurring networks, with the implicit convention that different networks depend on different subsets of these parameters.

We denote by $\mathrm{MLP}_\rho^\zeta[N_1, \ldots, N_T]$ a fully-connected network with layer sizes $N_1, \ldots, N_T$, the nonlinear activation function $\rho \colon \mathbb{R} \to \mathbb{R}$ after each layer, and parameters $\zeta$. For graph convolutional networks, we adapt the approach by Kipf & Welling (2017) to data $z_e^0 \in \mathbb{R}^{N_0}$ on edges $e$: The $t$th layer is given by

$$z_e^{t+1} = \rho \left( W_1^{t+1} z_e^t + W_2^{t+1} \sum_{\tilde{e} \in \mathcal{N}(e)} z_{\tilde{e}}^t + b^{t+1} \right) \tag{5}$$

where $W_1^{t+1} \in \mathbb{R}^{N_{t+1} \times N_t}$ and $W_2^{t+1} \in \mathbb{R}^{N_{t+1} \times N_t}$ are small matrices with learnable parameters and $b^{t+1} \in \mathbb{R}^{N_{t+1}}$ is the bias, all stored in the parameters $\zeta$. We define the neighborhood of an edge in a triangle mesh as $\mathcal{N}(e) := \{ \tilde{e} \in \mathcal{E} \mid e \text{ and } \tilde{e} \text{ share a vertex} \}$. We used the Exponential Linear Unit (Clevert et al., 2016) as activation function in all our experiments.

## 4 COMPOSITE NETWORK APPROXIMATION

As previously explained, we want to learn an efficient parametrization $\Phi \colon \mathbb{R}^m \to \mathcal{M}$ of an $m$-dimensional Riemannian data manifold $\mathcal{M}$. Below, we will explain our structural assumptions to achieve this, detail how we include them in the network architecture and training, and finally discuss their applicability to practical examples.

**Structural Assumptions**  One can think of the parametrization as the decoder part of an autoencoder, however, for the purpose of this article we want to be more specific: We would like the parametrization to have a geometric meaning, so we will focus on the situation in which $\Phi$ should encode the Riemannian exponential map at some point $z \in \mathcal{M}$. In principle, our approach can also be combined with alternative concepts in which no ground truth parametrization is available, but this setting allows a particularly simple quantification of the results.

If no additional structure of the data manifold is known, one can of course not improve over computing the Riemannian exponential directly or approximating it by some standard neural network. In contrast, we consider the case where a specific structure is known. For the purpose of our article, this structure is a priori given, that is, we do not study how such a structure can be found or obtained, though we give an example for a corresponding procedure on Riemannian shape spaces later on. We require that the data manifold $\mathcal{M}$ to be parametrized is embedded in some $\mathbb{R}^n$ for possibly large $n$. The structure of $\mathcal{M}$ we want to exploit has three components:

(0) **Correlation** We assume a structural correlation between the different coordinates, *e.g.* graph-neighbour relations for triangular meshes or pixel-neighbour relations for image data, so that convolutional networks can be applied.

(1) **Factorization** We assume that the manifold can be smoothly approximated by a product of much lower-dimensional manifolds $\mathcal{M}_1, \mathcal{M}_2, \ldots \mathcal{M}_J$, which we will parametrize separately. Thereby, we exploit that the necessary network size as well as the required training effort decrease with smaller manifold dimension: For $m$-dimensional manifolds the network size should scale at least linearly in $m$, while the required training set and thus also training time will scale exponentially in $m$.

(2) **Combination** It is not sufficient that the single factor manifolds are easy to parametrize since a generic point on $\mathcal{M}$ has components in all factors. Thus, we assume that the direct sum of all factors

$$\mathcal{M}_1 \oplus_z \ldots \oplus_z \mathcal{M}_J := \{z + (z_1 - z) + \ldots + (z_J - z) \,|\, z_1 \in \mathcal{M}_1, \ldots, z_J \in \mathcal{M}_J\}$$

for some $z \in \mathcal{M}$ already approximates the actual manifold $\mathcal{M}$ with a (possibly large, but) very smooth approximation error. This will ensure that a suitable map from $(z_1 - z, \ldots, z_J - z)$ to $\mathcal{M}$ can be efficiently learned.

The toy shape manifold $\mathcal{M}$ of tori from Figure 1 has the flat metric of $S^1 \times S^1 \times \mathbb{T}^2$ (the factors representing the orientation of the longitudinal and latitudinal ellipsoidal cross-sections as well as a bump position, see subsection 5.1) and thus satisfies condition (1). Condition (2) holds since each torus is represented as NRIC: For instance, the creation of the bump (which corresponds to changing the position in $\mathbb{T}^2$) is well described by simply adding fixed numbers in the right places of the edge length vector $l(X)$. However, the direct sum is indeed only an approximation – otherwise there would be no error in Figure 1 (green), but one clearly sees a remnant of the bump from the reference shape.

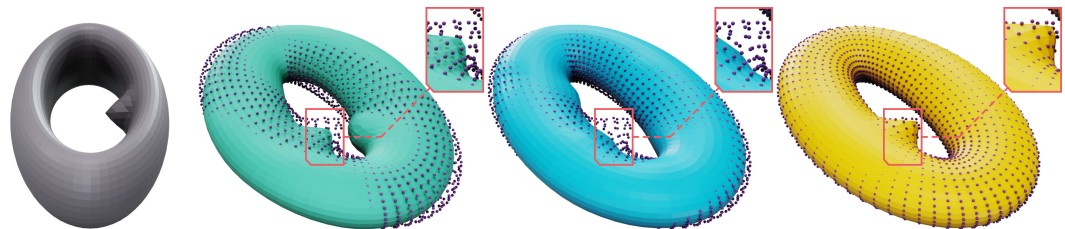

Figure 1: Approximation of the Freaky Torus. We show the reference shape $z$ in grey, the approximation of $\exp_z v$ by affine combination of exact factors in green, by a monolithic network in blue, and by our composite network in yellow. The correct vertex positions of $\exp_z v$ are shown as purple points. These purple points should ideally lie on the shaded surface, which would indicate a good fit. Indeed, for the approximation by our composite network, this is mostly the case, while for the other two approaches the approximation does not match the point cloud in many places.

**Network Representation and Training** Before we discuss the previous conditions, let us describe how we exploit them in our network architecture and training procedure. To learn the parametrization $\Phi \colon \mathbb{R}^m \to \mathcal{M}$, we decompose it as

$$\Phi = \Psi^\zeta \circ (\psi_1^\zeta, \ldots, \psi_J^\zeta),$$

where $\psi_j^\zeta \colon \mathbb{R}^{m_j} \to \mathbb{R}^n$ is the parametrization of the $m_j$-dimensional factor manifold $\mathcal{M}_j$ and $\Psi^\zeta \colon (\mathbb{R}^n)^J \to \mathbb{R}^n$ is the combination of the single factors, behaving approximately like $(x_1, \ldots, x_n) \mapsto z + (x_1 - z) + \ldots + (x_n - z)$, see Figure 2.

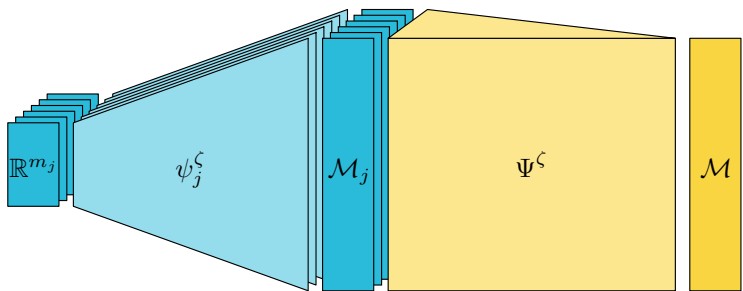

Figure 2: Structure of composite network.

The maps $\psi_j^\zeta$ will be fully connected neural networks. Because they approximate the smooth exponential map on rather low-dimensional spaces, they are expected to achieve a high approximation quality that is typically stable under variation of the concrete network architecture (number and size of layers). For the maps $\Psi^\zeta$, we exploit that they operate on a structured domain, *e.g.* a mesh, and thus they will be convolutional neural networks. This allows for efficient training and storage even though they operate on high-dimensional data. One could alternatively also use fully connected networks for $\Psi^\zeta$, but the observed quality of the results was similar despite significantly increased memory requirements to train, store, and evaluate.

These networks are trained separately: To train the map $\psi_j^\zeta$, we consider a parametrization

$$\omega_j \colon \mathbb{R}^{m_j} \to T_z\mathcal{M}, \quad \omega_j(a^j) = a_1^j e_1^j + \ldots + a_{m_j}^j e_{m_j}^j$$

with a coefficient vector $a^j = (a_1^j, \ldots, a_{m_j}^j)$ and a basis $e_1^j, \ldots, e_{m_j}^j$ of the Riemannian logarithm of $\mathcal{M}_j$ at point $z$ as a linear subspace of the tangent space $T_z\mathcal{M}$. We then consider a set of random samples $S^j \subset \mathbb{R}^{m_j}$ as training data (for instance normally distributed or uniformly on a ball) and minimize the loss function

$$\mathcal{J}_j(\zeta) = \frac{1}{|S^j|} \sum_{a^j \in S^j} \|\exp_z(\omega_j(a^j)) - \psi_j^\zeta(a^j)\|^2$$

in some norm $\|\cdot\|$ depending on the application. To train the map $\Psi^\zeta$, we subsequently consider a random training set $S \subset \mathbb{R}^{m_1} \times \ldots \times \mathbb{R}^{m_J} = \mathbb{R}^m$ and minimize the loss function

$$\mathcal{J}(\zeta) = \frac{1}{|S|} \sum_{(a^1, \ldots, a^J) \in S} \|\exp_z(\omega_1(a^1) + \ldots + \omega_J(a^J)) - \Psi^\zeta(\psi_1^\zeta(a^1), \ldots, \psi_J^\zeta(a^J)\|^2,$$

where we may or may not keep the maps $\psi_j^\zeta$ fixed.

**Applicability of Assumptions** While condition (0) essentially has to hold for any approach employing neural networks, conditions (1) and (2) are specific to our approach. Condition (1) expresses that the *intrinsic geometry* of the data manifold $\mathcal{M}$ has a simplifying structure, while condition (2) is about the *extrinsic geometry* of how $\mathcal{M}$ is embedded in $\mathbb{R}^n$ – only both conditions together characterize a structure that can efficiently be learned. Typical situations where our conditions hold include the following two examples:

- The different factor manifolds correspond to different spatial regions. For instance, images may sometimes be partitioned into different regions that can vary more or less independently of each other. If the regions are fully disjoint, the manifold $\mathcal{M}$ can exactly be written as a direct sum of factor manifolds; if on the other hand the regions slightly overlap, then this is only fulfilled approximately. Examples in shape spaces are where the single factor manifolds correspond to shape variations with disjoint support such as articulation of the different extremities of a character. Such a manifold structure is ubiquitous in computer graphics applications, and our computational examples mostly belong to this category.

- On shapes one can also often observe variations at different length scales that are independent of each other, for example, geometric texture on small length scales versus global shape variations at large length scales. The above toy example of the torus, in which the small bump is more or less independent of the global shape variations, belongs to this category.

As mentioned before, we assume the product manifold structure to be given a priori (in the torus example, for instance, it was given by design of the data manifold). The product manifold structure would typically be a result of some data manifold analysis, for example from disentanglement learning. In the setting of Riemannian shape spaces, one way to obtain this structure is the SPGA introduced earlier: It makes use of the decomposition into independent product manifolds due to disjoint support of the corresponding shape variations.

## 5 EXPERIMENTS AND APPLICATIONS

In this section, we present experimental results on the aforementioned synthetic shape manifold of deformed tori and manifolds extracted via SPGA. We will compare our method to approaches based on (Sassen et al., 2020b) and a straightforward approximation of the parametrization by a single fully connected network. To quantify the approximation quality of different approaches, we use the coefficient of determination $R^2$. For approximations $\tilde{z}_i$ of NRIC $z_i$ with mean $\bar{z}$, it is defined as

$$R^2(z, \tilde{z}) = 1 - \frac{\sum_i \|\tilde{z}_i - z_i\|_{2,\bar{z}}^2}{\sum_i \|z_i - \bar{z}\|_{2,\bar{z}}^2}.$$

From a statistical point of view, it quantifies the proportion of variation of the data that is explainable by a given model. This means an $R^2$ of one is optimal, the smaller it is the worse is the approximation, and a negative $R^2$ means that the model is worse than simply using the mean.

**Training & Implementation**    We used Adam (Kingma & Ba, 2015) as descent method for training all networks, where the initial learning rate was $10^{-3}$ and was reduced by a factor of 10 every time the loss did not decrease for multiple iterations. For regularization, we used batch normalization after each layer and a moderate dropout regularization ($p = 0.1$) after each convolutional layer. We implemented the neural networks in PyTorch (Paszke et al., 2019) using the PyTorch Geometric library (Fey & Lenssen, 2019). The tools for the NRIC manifold were implemented in C++ based on OpenMesh (Botsch et al., 2002), where we use the Eigen library (Guennebaud et al., 2010) for numerical linear algebra. We follow an approach similar to Kilian et al. (2007) to perform all computations on a coarsened mesh and prolongate solutions to a fine one only for visualizations.

### 5.1 SYNTHETIC DATA: FREAKY TORUS

For the *Freaky Torus* dataset, we construct a synthetic shape space with factors $S^1 \times S^1 \times \mathbb{T}^2$, where $\mathbb{T}^2$ refers to the flat 2-dimensional torus. It is realized in NRIC by (i) deforming the two cross-sectional circles of a torus to ellipses of fixed aspect ratio and orientation controlled by the first two $S^1$ factors and (ii) growing a bump in normal direction whose position is controlled by the last $\mathbb{T}^2$ factor. These torus deformations are applied to a regular mesh of a regular torus embedded in $\mathbb{R}^3$, and the deformed meshes' NRIC are extracted to obtain our datapoints. We used a mesh with 2048 vertices and uniformly drew 1000 samples from $S^1 \times S^1 \times \mathbb{T}^2$.[1] More details on the data generation can be found in Appendix A.

Figure 1 shows that a single fully connected network struggles with approximating the high frequency detail of the bump, while our composite network is able to handle this well. This can also be observed in the approximation quality quantified using the $R^2$. The composite network achieves an $R^2$ of 0.99 and the monolithic network one of 0.95. This difference may sound small, however, this is because the bump is a detail and the error is dominated by the overall shape of the torus.

### 5.2 APPLICATION: SPGA MANIFOLDS

Lastly, we report the results of applying our method to shape manifolds whose approximate product structure is found with the help of SPGA. To this end, we repeat three of the examples discussed by Sassen et al. (2020b) and consider one new dataset. The repeated examples are a humanoid dataset from (Anguelov et al., 2005), a dataset of face meshes from (Zhang et al., 2004), and a set of hand meshes from (Yeh et al., 2011). For the new example, we examine a humanoid dataset based on SMPL-X (Pavlakos et al., 2019), where we consider their *expressive hands and faces* (EHF) dataset,

---

[1]The data and the generating code can be found at `https://gitlab.com/jrsassen/freaky-torus`.

containing 100 shapes, and 49 additional shapes from the SMPL+H dataset, which feature more expressive arm and leg movements, adding to a total of 149 input shapes. We interpret all shapes as elements of the nonlinear NRIC space so that this small amount of data points already suffices to span a high-dimensional, nonlinear NRIC submanifold that serves as our data manifold.

Based on this data, we follow the numerical approach of Sassen et al. (2020b) to solve (1) and thereby compute the sparse tangent modes. We report the chosen number $m$ of included modes in Table 1, where we used the same number as Sassen et al. (2020b) for the repeated examples. To factor the resulting data manifold, we again follow the approach outlined by Sassen et al. (2020b), which is a clustering based on the spatial overlap of the modes. For the hand and face examples, we choose the same number of factors $J$, while for the SCAPE example we decreased the number to account for the possibility to handle higher-dimensional factors with our method. Our choices are again documented in Table 1. Each cluster then spans exactly one of the factor manifolds and the range of their dimensions is also reported in said table.

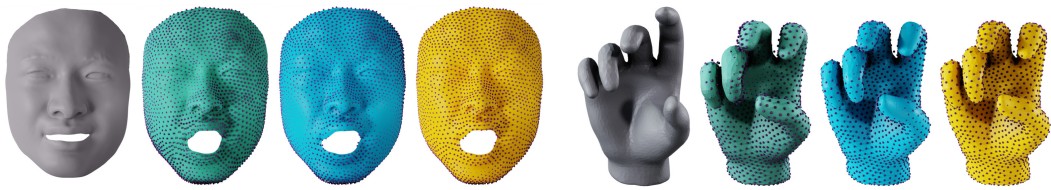

Figure 3: Examples from the face and hands datasets. We use the same colors as in Figure 1.

**Data Generation** Next, we sample the exponential map on the space spanned by the SPGA modes to generate the training (and test) data. To this end, we consider the hypercube in $\mathbb{R}^m$ given by the minimal and maximal coefficients of projections of the input data onto the SPGA subspace. Then we draw our parametrization coefficient samples $S \subset \mathbb{R}^m$ uniformly from this hypercube. To create the samples $S^j$ for the factor manifolds, we simply take the corresponding subcomponents of coefficient vectors from $S$. The corresponding shapes were then computed by evaluating the exponential map for each of them. Overall, we sampled approximately $|S| = 4000$ points for each of the considered examples. The dataset was split randomly into a training (80 %) and a test (20 %) set, with the training set being used for the descent method of the loss functionals and the test set being used to evaluate the performance of the networks.

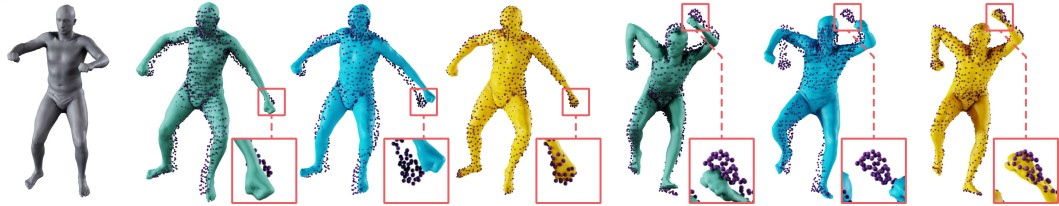

Figure 4: Two examples from the SCAPE dataset. We use the same colors as in Figure 1.

| Example | $m$ | $J$ | $m_j$ | Affine | Monolithic | Composite (Ours) | Sassen et al. (2020b) |
|---|---|---|---|---|---|---|---|
| SMPL+X | 80 | 10 | 3 − 24 | 0.78 | 0.85 | 0.93 | — |
| SCAPE | 40 | 6 | 5 − 9 | 0.77 | 0.60 | 0.91 | — |
| Hands | 12 | 4 | 2 − 4 | 0.88 | 0.95 | 0.98 | 0.80 |
| Faces | 10 | 6 | 1 − 4 | 0.96 | 0.95 | 0.99 | 0.95 |

Table 1: Approximation quality $R^2$ on SPGA examples.

**Comparison to Affine Approximation** Sassen et al. (2020b) also proposed a scheme based on multilinear interpolation of precomputed exponentials for each of the factor manifolds and subsequent affine combination of the results to approximate the exponential map and parametrize $\mathcal{M}$. A natural question is how our network-based approach compares to this.

For the humanoid examples, it was not possible to precompute Riemannian exponentials on a regular grid for all factor manifolds due to their high dimensionality. Hence, instead of multilinearly interpolating precomputed exponentials we simply compute the exponentials within each factor manifold exactly before combining them affinely. We dub this method simply 'affine combination' and present its results in Figure 4 and Table 1; it is computationally heavy, but yields an upper bound on the quality of the method by Sassen et al. (2020b). The limitation does not apply to our new approach, which for example allows us to learn an efficient parametrization for the SMPL+X dataset, where the expressive movements of hand and face require a higher-dimensional data manifold.

In all examples, our composite network approach achieves higher approximation accuracy than the 'affine combination'. This shows that the network $\Psi^\zeta$ is able to correct the approximation errors of the direct sum structure. For the lower-dimensional examples, this difference is not as pronounced since their sparse modes have a better support separation.

Furthermore, storing our network-based approximation requires less memory than the approach by Sassen et al. (2020b). For example, on the SCAPE dataset storing grids with approx. 20000 samples as reported by Sassen et al. (2020b) requires about $1.7\,\mathrm{GB}$ of storage, while our networks only require $0.6\,\mathrm{GB}$ (without optimizing for a small memory footprint).

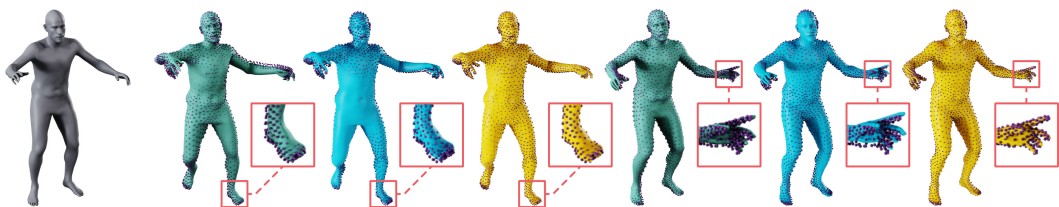

Figure 5: Two examples from the SMPL-X dataset. We use the same colors as in Figure 1.

**Comparison to Monolithic Network**  Another obvious question is whether our composite approach shows any benefit over training a simple, single network. For evaluation, we also trained one fully connected network $\widetilde{\Phi}^\zeta \colon \mathbb{R}^m \to \mathcal{M}$ to approximate the parametrization at once, dubbed 'monolithic' approach. The corresponding approximation qualities are reported in Table 1. One sees that for the lower-dimensional examples this monolithic approach achieves an approximation quality close to the one of our composite network. However, for the higher-dimensional, humanoid examples the approximation quality of the monolithic approach is noticeably lower.

# 6 CONCLUSION

Our results suggest that the most fundamental geometric operation on Riemannian data manifolds, the parametrization of the manifold via the Riemannian exponential map, can in principle be learned. We illustrated this on shape manifolds of triangular meshes, for which the exponential map is computationally expensive so that an approximation is attractive. However, while naive implementations via deep neural networks proved[7] ineffective, we achieved consistently satisfying results by matching our training and network architecture to a typical structure of shape manifolds: that they can be approximated by an affine sum of submanifolds. We thus learned both, the lower-dimensional Riemannian exponential map on each submanifold as well as the (close to affine) composition of the different submanifolds. We furthermore illustrated in our examples that such manifold structures arise from basic principles like support or scale separation of different shape variations.

While we implemented the above concept of composite networks only for shape manifolds, it should also be applicable to image manifolds. However, first the corresponding tools to identify approximate product manifold structures (such as the SPGA) would have to be developed for images. It is also conceivable to replace the SPGA by learning approaches akin to disentanglement learning. Furthermore, we did not touch upon further optimization for the specific setting of our shape manifolds: The single factor manifolds typically describe localized shape variations so that a sparsity regularization of the corresponding networks would make sense and could substantially reduce the parameter size. Furthermore, one could add a regularization favoring those NRIC that correspond to an immersed mesh, thereby reducing postprocessing.

ACKNOWLEDGMENTS

This work was supported by the Deutsche Forschungsgemeinschaft (DFG, German Research Foundation) via project 211504053 – Collaborative Research Center 1060, project 212212052 – "Geodesic Paths in Shape Space" (part of the NFN Geometry + Simulation), and via Germany's Excellence Strategy project 390685813 – Hausdorff Center for Mathematics and project 390685587 – Mathematics Münster: Dynamics–Geometry–Structure.

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

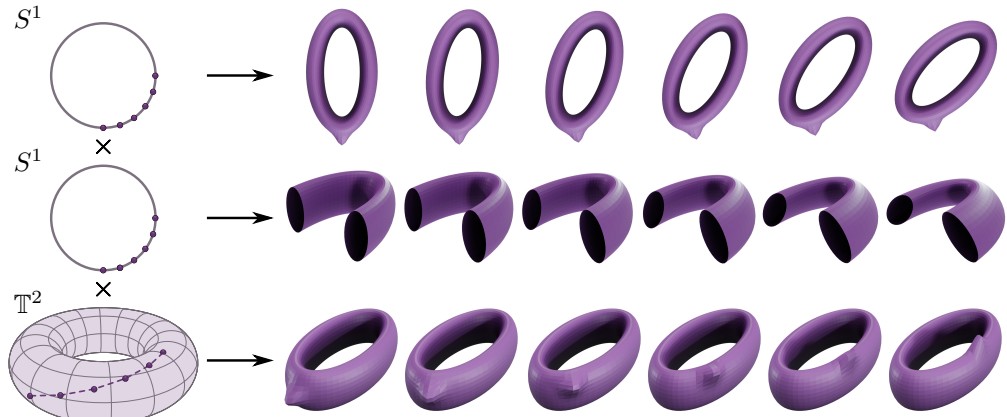

Figure 6: Factors of the *Freaky Torus*. We visualize our synthetic shape space by demonstrating the effect of moving along the individual factors to the final shape. In the first row, we see how the first factor, an $S^1$, controls the deformation of the latitudinal cross-section. We show the deformed torus from the top. In the second row, we see how the next factor, another $S^1$, controls the deformation of the longitudinal cross-section. Here, we show the torus cut in half to better highlight the cross-section's shape. In the last row, we show how the third factor, a two-dimensional flat torus $\mathbb{T}^2$, controls the position of a bump on the deformed torus.

## A    FREAKY TORUS

In this appendix, we provide more details on the construction of our shape space *Freaky Torus* of deformed tori, a synthetic shape space with factors $S^1 \times S^1 \times \mathbb{T}^2$, whose action is also summarized in Figure 6. We will first derive its continuous version leading to a family of parametrization and then arrive at the discrete version via a simple spatial discretization. To begin, we recall the parametrization $f \colon [0, 2\pi)^2 \to \mathbb{R}^3$ of a standard torus of revolution with radii $R$ and $r$ of the latitudinal and longitudinal circular cross-sections respectively, which is given by

$$f(u,v) := R \begin{pmatrix} \cos(u) \\ \sin(u) \\ 0 \end{pmatrix} + r \begin{pmatrix} \cos(u)\,\cos(v) \\ \sin(u)\,\cos(v) \\ \sin(v) \end{pmatrix}. \tag{6}$$

The first two factors $S^1 \times S^1$ of our shape space control the deformation of these cross-sections into ellipses and hence we want to replace the parametrizations of the cross-sectional circles by ones of appropriate ellipses. To this end, a circle deformed into an ellipse with semi-axes' lengths $a$ and $b$ rotated by $\eta/2$ against the coordinate axis is parametrized by

$$\tau^{\eta,a,b}(t) := \begin{pmatrix} a\cos(\frac{\eta}{2})\cos(t - \frac{\eta}{2}) - b\sin(\frac{\eta}{2})\sin(t - \frac{\eta}{2}) \\ a\sin(\frac{\eta}{2})\cos(t - \frac{\eta}{2}) + b\cos(\frac{\eta}{2})\sin(t - \frac{\eta}{2}) \end{pmatrix}. \tag{7}$$

The phase-shift in the parametrization comes from the fact that the rotation of the semi-axes is also achieved by a warping of the circle instead of rotating it. We introduced this parametrization with the half-scaling of $\eta$ for cosmetic reasons so that later all deformations are parametrized over the same interval $[0, 2\pi)$.

Now, we use this parametrization as replacement for the circular cross-section in the parametrization $f$ of the torus. We fix the semi-axes' lengths $a_R, b_R$ and $a_r, b_r$ of the latitudinal and longitudinal cross-sections respectively and introduce their rotation as parameters $\alpha, \beta \in [0, 2\pi)$. This leads to the parametrization

$$f^{\alpha,\beta}(u,v) := R \begin{pmatrix} \tau_1^{\alpha,a_R,b_R}(u) \\ \tau_2^{\alpha,a_R,b_R}(u) \\ 0 \end{pmatrix} + r \begin{pmatrix} \cos(u)\,\tau_1^{\beta,a_r,b_r}(v) \\ \sin(u)\,\tau_1^{\beta,a_r,b_r}(v) \\ \tau_2^{\beta,a_r,b_r}(v) \end{pmatrix}. \tag{8}$$

The last factor $\mathbb{T}^2$ controls the position on a bump on such a torus. To describe the corresponding deformation, we first need the normal of the surface in which direction the bump will point. It is

given by the usual formula for smooth surfaces namely

$$n^{\alpha,\beta}(u,v) := \frac{\partial_u f^{\alpha,\beta} \times \partial_v f^{\alpha,\beta}}{\|\partial_u f^{\alpha,\beta} \times \partial_v f^{\alpha,\beta}\|}(u,v). \tag{9}$$

Then the bump is a deformation in the direction of this normal around the position determined by $(\gamma,\zeta) \in [0,2\pi)^2$ with maximal height $h$. To limit the support of the deformation, we use a simple Gaussian with parameter $\varepsilon$ on the distance to the center point.

We finally arrive at the parametrization of the continuous version of our synthetic shape space

$$\mathcal{F}: [0,2\pi) \times [0,2\pi) \times [0,2\pi)^2 \to \left([0,2\pi)^2 \to \mathbb{R}^3\right)$$
$$(\alpha,\beta,\gamma,\zeta) \mapsto \left((u,v) \mapsto f^{\alpha,\beta}(u,v) + h\, e^{-\frac{\|f^{\alpha,\beta}(u,v)-f^{\alpha,\beta}(\gamma,\zeta)\|^2}{\varepsilon^2}} n^{\alpha,\beta}(\gamma,\zeta)\right), \tag{10}$$

where the images are parametrizations of embedded surfaces. To obtain discrete surfaces, we apply these to a fixed triangulation of the torus, which yields the discrete version of the shape space.

For the dataset we used in our experiments, we chose $a_R = a_r = 1, b_R = b_r = \frac{1}{2}, R = 0.375, r = 0.125, h = 0.075$, and $\varepsilon = 0.05$. It is available along with the code at `https://gitlab.com/jrsassen/freaky-torus`.

## B  ADDITIONAL RESULTS

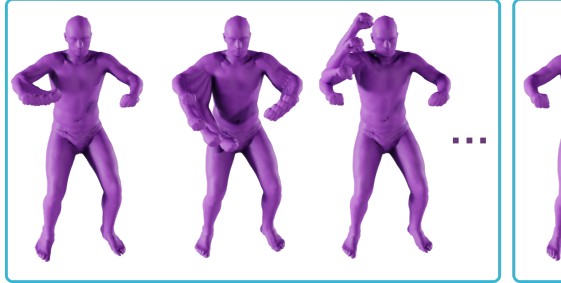
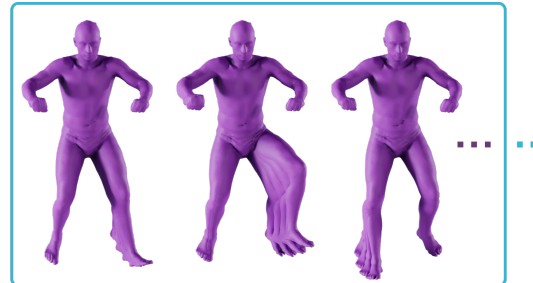

Figure 7: Exemplary SPGA modes extracted from the SCAPE dataset. We show a selection of modes generating two of the factor manifolds (indicated by the blue frames; not all modes of each factor manifold are shown as indicated by the purple dots). Note that while these deformations move many nodal positions, their support in NRIC is indeed localized. For example, the fifth mode is supported mainly in the hip region of the shape. This also links it with the other modes in the same group even though they might move a different leg.

**Sparsity**  In Section 4, we postulated several assumptions on the structure of our data manifold, which raises the question if they are actually satisfied in our experiments. Since we work with NRIC, assumption (0) is fulfilled as they are given as data on the edges of a mesh (see also Appendix D). Then, for the freaky torus example, we explicitly constructed a product manifold (hence assumption (1) is fulfilled) and chose the factors such that they act on different length scales, which entails the fulfillment of assumption (2). For the examples using a decomposition via SPGA, we rely on it producing sparsely supported modes that can be grouped by their spatial overlap. As already Sassen et al. (2020b) observed, the resulting factorization of the data manifold fulfills our assumptions. In Figure 7, we show exemplary modes from the SCAPE dataset highlighting that this is indeed the case.

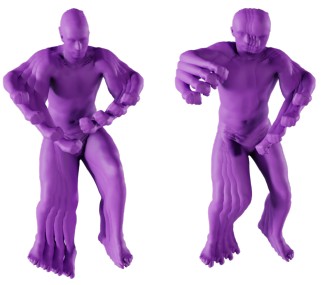

Figure 8: Exemplary PGA modes extracted from the SCAPE dataset showing the global support typical for such modes.

The sparsity of the SPGA modes is a result of the regularization $\mathcal{R}$ in the SPGA problem (1). To still achieve a good approximation of the input data, the modes

typically localize in different regions of the shape. This allows us to group them as explained before to obtain the factor manifolds that fulfill assumptions (1) and (2). In contrast, this is not the case for PGA modes, *i.e.* if we do not use any regularization there is no reason to expect localized support, which is also illustrated in Figure 8. This means our method is not applicable for such modes.

**Animation**   One possible application of our composite network is efficient animation of shapes. In this context, we can consider shape interpolation and extrapolation problems, which correspond to the evaluation of the Riemannian logarithm and exponential map as explained in Section 3 and Appendix C. For the case of shape interpolation, we are given two shapes by their latent coordinates $a(0) \in \mathbb{R}^m$ and $a(1) \in \mathbb{R}^m$ respectively. Then, the latent coordinates of intermediate shapes are obtained by linear interpolation, *i.e.* we define $a(t) \coloneqq t\, a(1) + (1-t)a(0)$ for $t \in [0, 1]$. By evaluating our composite network on these coordinates, we obtain the approximate NRIC $z(t) \coloneqq \Psi^\zeta(\psi_1^\zeta(a^1(t)), \ldots, \psi_J^\zeta(a^J(t)))$ of these shapes, where $a^j(t) \in \mathbb{R}^{m_j}$ are the factorized coordinates as before. This leads to a smooth interpolation between shapes as demonstrated in Figure 9 and the supplementary video. Shape extrapolation can be equivalently phrased by considering linear extrapolation in the latent space $\mathbb{R}^m$.

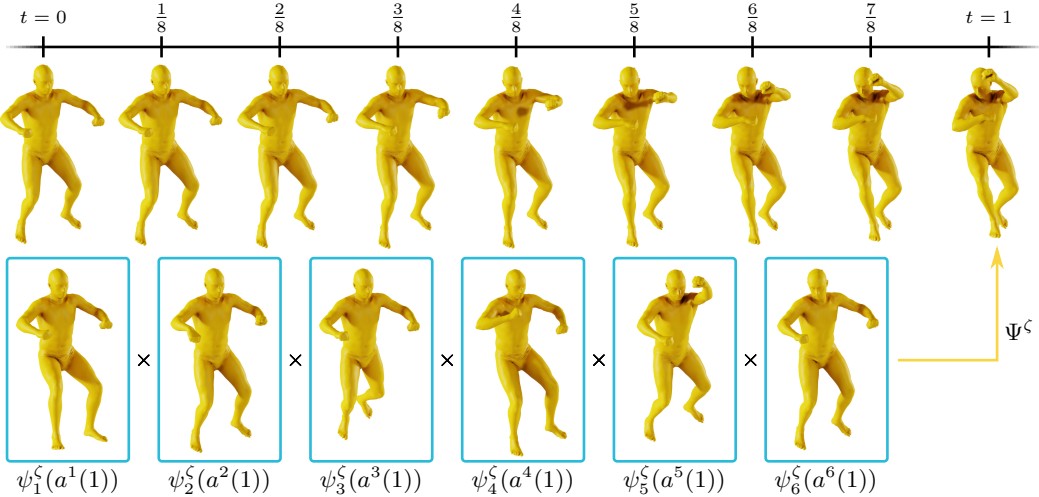

Figure 9:   For two given shapes with latent coordinates $a(0)$ and $a(1)$, we compute interpolating NRIC $z(t)$ using our composite network. In the top row, we see the surfaces reconstructed from these NRIC for intermediate time steps exhibiting smooth deformations. Below, we also show the elements $\psi_j^\zeta(a^j(1))$ from the factor manifolds $\mathcal{M}_j$ which lead to the final shape by applying the combination network $\Psi^\zeta$. These individual factors lead primarily to deformations of the legs for $\psi_1$ and $\psi_3$, of the arms for $\psi_4$ and $\psi_5$, of the wrists for $\psi_2$, and of the head for $\psi_6$. See also the supplementary video.

**Number of Samples**   We observed that our composite network can also be trained with smaller amounts of samples than we used in subsection 5.2. For example on the SCAPE dataset, if we only use 20 % of the data as training set (about 800 samples) then we still achieve an $R^2$ of 0.86. Even if we use a mere 5 % (200 samples) we still reach an $R^2$ of 0.77. We observed a similar behavior on the SMPL+X dataset with an $R^2$ of 0.85 at 20 % training data and of 0.74 at 5 % training data.

**Runtimes**   Our network-based approach enables runtime efficient approximation of the exponential map. For example, on the SCAPE dataset, we used $K = 16$ time steps to evaluate the time-discrete exponential map (see Appendix C) when generating the training samples. The computation for each such evaluation required around 8 seconds. In contrast, evaluating the networks takes about 10 milliseconds. To render the result, we have to reconstruct the nodal positions of the triangle mesh from the NRIC, for which we use the nonlinear least-squares method from Fröhlich & Botsch (2011). This requires a small number (*e.g.* 2 to 3 in Figure 9) of Gauß–Newton iterations taking about 20ms each. Overall the performance is comparable to the approach by Sassen et al. (2020b), which in contrast to our approach is limited in the amount of latent dimensions it can handle, though.

## C    RIEMANNIAN OPERATORS AND THEIR DISCRETIZATIONS IN BRIEF

The tangent space $T_z\mathcal{S}$ of a manifold $\mathcal{S}$ in the point $z \in \mathcal{S}$ is the vector space of all velocities a path in $\mathcal{S}$ can have when passing through $z$. A Riemannian manifold equips each of these tangent spaces $T_z\mathcal{S}$ with an inner product $g_z(\cdot, \cdot)$ so that norms of velocity vectors and angles between them can be measured. The length of a path $\gamma : [0, 1] \to \mathcal{S}$ in the Riemannian manifold then is the time integral of the norm of its velocity $\dot{\gamma}$, $\mathcal{L}[\gamma] = \int_0^1 \sqrt{g_{\gamma(t)}(\dot{\gamma}(t), \dot{\gamma}(t))} \, \mathrm{d}t$. Given two points $z_0, z_1 \in \mathcal{S}$, the shortest connecting path $\gamma$ with $\gamma(0) = z_0$, $\gamma(1) = z_1$ is called a geodesic, and the Riemannian distance $\mathrm{dist}(z_0, z_1)$ between both points is defined as its length. It is essentially an application of Jensen's inequality that the geodesic connecting $z_0$ and $z_1$ can equivalently be found by minimizing the path energy

$$\mathcal{E}[\gamma] = \int_0^1 g_{\gamma(t)}(\dot{\gamma}(t), \dot{\gamma}(t)) \, \mathrm{d}t$$

instead of $\mathcal{L}[\gamma]$. Yet another view of geodesics, following from the optimality conditions of the above minimization, is that they are paths that always go straight and at constant speed, *i.e.* they do not accelerate (neither do they change the motion direction, nor do they change the velocity along this direction; on the earth's surface, for instance, geodesics are great circles). This last viewpoint suggests to associate to every $v$ in the tangent space $T_z\mathcal{S}$ at an arbitrary $z \in \mathcal{S}$ a point $y \in \mathcal{S}$, which is defined as follows: Given $v \in T_z\mathcal{S}$, just start out at $z$ with initial velocity $v$ and continue straight (*i.e.* along a geodesic) for total time $1$. The map which maps $v$ to the arrival point $y$ is the so-called Riemannian exponential map

$$\exp_z : T_z\mathcal{S} \to \mathcal{S}, \qquad \exp_z v = y.$$

Via this exponential map one can identify the tangent space $T_z\mathcal{S}$ with the manifold $\mathcal{S}$ (at least if $\exp_z$ is injective, otherwise $T_z\mathcal{S}$ can be identified with a multiple covering of $\mathcal{S}$). The inverse map is the Riemannian logarithm $\log_z : U \to T_z\mathcal{S}$, defined for a small enough neighborhood $U \subset \mathcal{S}$ of $z$. For a point $z_1 \in U$, it tells us the initial velocity $\log_z z_1$ of the geodesic from $z$ to $z_1$.

Riemannian geodesics, logarithms and exponentials are expensive to approximate computationally. The computation typically has to be performed in charts or local parametrizations of the manifold, *i.e.* $\mathcal{S}$ is identified with (an open subset of) $\mathbb{R}^n$ and the Riemannian metric with a symmetric positive definite matrix $G_z \in \mathbb{R}^{n \times n}$ depending on $z \in \mathbb{R}^n$. A variational discretization by Rumpf & Wirth (2015) approximates continuous paths $\gamma$ by time-discrete paths $(\gamma_0, \ldots, \gamma_K)$ to be interpreted as polygonal paths in $\mathbb{R}^n$ with vertices $\gamma_0, \ldots, \gamma_K$ at times $\frac{0}{K}, \frac{1}{K}, \ldots, \frac{K}{K}$. The path energy is then approximated by a discrete path energy

$$E[(\gamma_0, \ldots, \gamma_K)] = K \sum_{k=1}^{K} W(\gamma_{k-1}, \gamma_k),$$

where $W(z_0, z_1)$ is a second order accurate approximation to the squared Riemannian distance $\mathrm{dist}^2(z_0, z_1)$ (for instance, $W(z_0, z_1) = (z_0 - z_1)^T G_{z_0}(z_0 - z_1)$). $E$ may be viewed as a Riemann sum approximation of the integral in $\mathcal{E}$. Minimizing $E$ under fixed end points then yields a discrete $K$-geodesic, *i.e.* a discrete approximation $(\gamma_0, \ldots, \gamma_K)$ to a geodesic between $\gamma_0$ and $\gamma_K$. The initial velocity $K(\gamma_1 - \gamma_0)$ of this polygonal path is the discrete approximation of the Riemannian logarithm $\log_{\gamma_0} \gamma_K$. The discretization of the Riemannian exponential works the other way round: Given a velocity vector $v \in \mathbb{R}^n$ we approximate $\exp_{\gamma_0} v \in \mathbb{R}^n$ by that point $\gamma_K \in \mathbb{R}^n$ such that the discrete $K$-geodesic $(\gamma_0, \ldots, \gamma_K)$ has initial velocity $K(\gamma_1 - \gamma_0) = v$. This point can be found by a time stepping procedure — one first sets $\gamma_1 = \gamma_0 + \frac{v}{K}$ and then iteratively computes $\gamma_2, \gamma_3, \ldots \gamma_K$ as follows: Since each triplet $\gamma_{k-1}, \gamma_k, \gamma_{k+1}$ of a discrete $K$-geodesic forms a discrete 3-geodesic (much like any subsegment of a continuous geodesic is itself again a geodesic), $\gamma_k$ must minimize $W[\gamma_{k-1}, \gamma_k] + W[\gamma_k, \gamma_{k+1}]$. The corresponding nonlinear optimality condition

$$0 = \partial_2 W[\gamma_{k-1}, \gamma_k] + \partial_1 W[\gamma_k, \gamma_{k+1}]$$

is then solved via Newton's method for $\gamma_{k+1}$, given $\gamma_{k-1}$ and $\gamma_k$.

Instead of working on charts, one can also consider an implicitly defined manifold $\mathcal{M} = \{z \in \mathbb{R}^m \,|\, \mathcal{Q}(z) = 0\}$, for suitable smooth functions $\mathcal{Q} : \mathbb{R}^m \to \mathbb{R}^r$. In this case, one proceeds analogously constraining the search for points on the manifold via a Lagrangian approach.

## D    A RECAP OF NONLINEAR ROTATION-INVARIANT COORDINATES

Let us briefly review the tools introduced by Wang et al. (2012) for a discrete version of the fundamental theorem of surfaces. Consider a simply connected, triangular surface with the set of vertices $\mathcal{V}$, edges $\mathcal{E} \subset \mathcal{V} \times \mathcal{V}$, and faces $\mathcal{F} \subset \mathcal{V} \times \mathcal{V} \times \mathcal{V}$. For a given vector of vertex positions $X \in \mathbb{R}^{3|\mathcal{V}|}$, we introduce the vector of all edge lengths $l(X) = (l_e(X))_{e \in \mathcal{E}}$ and the vector of all dihedral angles $\theta(X) = (\theta_e(X))_{e \in \mathcal{E}}$. As discussed by Wang et al. (2012), to ensure that the edge length and dihedral angle data $z = (l, \theta) \in \mathbb{R}^{2|\mathcal{E}|}$ actually corresponds to a triangular surface immersed in $\mathbb{R}^3$, two admissibility conditions have to be fulfilled.

The obvious first condition is the triangle inequality for the edge lengths on all triangles. We write this condition in formulas as $\mathcal{T}_f(l) > 0$ for all $f \in \mathcal{F}$ where $\mathcal{T}_f(l) = (l_i + l_j - l_k \quad l_i - l_j + l_k \quad -l_i + l_j + l_k)$ for a face $f \in \mathcal{F}$ with edge lengths $l_i, l_j, l_k$, and the above inequality is meant componentwise. Fulfillment of these conditions guarantees that we can construct individual triangles from given edge lengths. However, we need a second condition assuring that these triangles fit together with the given dihedral angles to form a surface, *i.e.* to guarantee integrability of $z$. For simply-connected discrete surfaces, this can be broken down to individual conditions for the fans of triangles surrounding any vertex $v$ in the set of interior vertices $\mathcal{V}_0$. Formally, we express this individual condition as $\mathcal{Q}_v(z) = 0$, which guarantees that we can construct the geometry of this fan from $z$. The explicit formula for this was introduced by Wang et al. (2012). If this condition is fulfilled for all interior vertices, one can show that it is indeed possible to construct the geometry of the entire surface. Sassen et al. (2020a) demonstrated that $\mathcal{Q}_v$ and its derivatives can be robustly and efficiently computed using quaternions. The conditions can be extended to higher-genus surfaces by including integrability conditions along non-contractible paths that generate the fundamental group on the triangular surface, but this is not used here.

The manifold of all $z \in \mathbb{R}^{2|\mathcal{E}|}$ corresponding to immersed triangular surfaces of the given mesh connectivity can be given by

$$\mathcal{M} = \big\{ z \in \mathbb{R}^{2|\mathcal{E}|} \,\big|\, \mathcal{T}(z) > 0, \, \mathcal{Q}(z) = 0 \big\},$$

where we collect all constraints in vector-valued functionals $\mathcal{T} = (\mathcal{T}_f)_{f \in \mathcal{F}}$ and $\mathcal{Q} = (\mathcal{Q}_v)_{v \in \mathcal{V}_0}$. As Sassen et al. (2020a), we call the manifold $\mathcal{M}$ the NRIC manifold (Nonlinear Rotation-Invariant Coordinates). The differential structure of this manifold is at first described in terms of the tangent space, which is given at position $z \in \mathcal{M}$ by $T_z\mathcal{M} = \ker D\mathcal{Q}(z) := \{w \in \mathbb{R}^{2|\mathcal{E}|} \,|\, D\mathcal{Q}(z)w = 0\}$. Here the matrix $D\mathcal{Q}(z) \in \mathbb{R}^{3|\mathcal{V}_0| \times 2|\mathcal{E}|}$ is the Jacobian of $\mathcal{Q}$. The triangle inequalities define an open set of $\mathbb{R}^{2|\mathcal{E}|}$ and are thus not needed to define the tangent space.

The advantage of NRIC is that they allow a local description of shell deformations based on the local variation of the edge lengths, which encodes membrane distortions, and the local variation of the dihedral angles, which encodes bending distortions. We use a Riemannian metric $g$ on the manifold $\mathcal{M}$ that reflects the physical dissipation caused by these infinitesimal variations of the discrete surface. In detail, the metric coincides with the Hessian of an elastic energy $W$, *i.e.* $g_z \colon \mathbb{R}^{2|\mathcal{E}|} \times \mathbb{R}^{2|\mathcal{E}|} \to \mathbb{R}$ with $g_z = \frac{1}{2}\mathrm{Hess}W[z, \cdot]$ restricted to $T_z\mathcal{M} \times T_z\mathcal{M}$. We use an elastic energy describing a deformation from a configuration $z$ to a configuration $\tilde{z}$ that decomposes into a membrane energy and a bending energy, *i.e.* $W[z, \tilde{z}] = W_{\mathrm{mem}}[z, \tilde{z}] + W_{\mathrm{bend}}[z, \tilde{z}]$. The bending energy is given by $W_{\mathrm{bend}}[z, \tilde{z}] = \sum_{e \in \mathcal{E}}(\theta_e - \tilde{\theta}_e)^2 d_e^{-1} l_e^2$, where $d_e = \frac{1}{3}(a_f + a_{f'})$ for the two faces $f$ and $f'$ adjacent to $e \in \mathcal{E}$ ($a_f$ is the area of $f$). Furthermore, the membrane energy is given by $W_{\mathrm{mem}}[z, \tilde{z}] = \sum_{f \in \mathcal{F}} a_f \cdot W_{\mathrm{mem}}(\mathrm{G}[z, \tilde{z}]|_f)$, where $W_{\mathrm{mem}}(A) := \frac{\mu}{2}\mathrm{tr}\, A + \frac{\lambda}{4}\det A - \big(\mu + \frac{\lambda}{2}\big)\log\det A - \mu - \frac{\lambda}{4}$. The constants $\mu$ and $\lambda$ are positive material constants, and $\mathrm{G}[z, \tilde{z}]$ denotes the Cauchy–Green strain tensor of the deformation — describing the face-wise distortion — as a function of the edge lengths on each face. Let us remark that the logarithmic term in the energy density $W_{\mathrm{mem}}$ acts as a barrier ensuring the triangle inequalities for finite-energy deformations.

# E  LIST OF SYMBOLS

**Riemannian Shape Spaces (see also Appendix C)**

| | |
|---|---|
| $\mathcal{S}$ | Shape space, p. 3 |
| $z$ | Point of a shape space, p. 3 |
| $T_z\mathcal{S}$ | Tangent space of $\mathcal{S}$ at point $z$, p. 3 |
| $g_z$ | Riemannian metric at point $z$, p. 3 |
| $\log_z$ | Riemannian logarithmic map at point $z$, p. 3 |
| $\exp_z$ | Riemannian exponential map at point $z$, p. 3 |

**Principal Geodesic Analysis**

| | |
|---|---|
| $\bar{z}$ | Riemannian center of mass, p. 3 |
| $\mathcal{R}$ | Regularization functional, p. 3 |

**Triangle Meshes and NRIC (see also Appendix D)**

| | |
|---|---|
| $\mathcal{V}$ | Vertices of a triangular surface, p. 3 |
| $\mathcal{E}$ | Edges of a triangular surface, p. 3 |
| $\mathcal{F}$ | Triangles of a triangular surface, p. 3 |
| $\mathcal{N}(e)$ | Neighbors of an edge in a triangle mesh, p. 4 |
| $X$ | Vector storing the vertex positions of all vertices of a triangular surface, p. 3 |
| $l$ | Vector of edge lengths, p. 3 |
| $l_e$ | Length of edge $e$, p. 3 |
| $\theta$ | Vector of dihedral angles, p. 3 |
| $\theta_e$ | Dihedral angle of triangles meeting at edge $e$, p. 3 |
| $\|\cdot\|_{p,\bar{z}}$ | Weighted $L^p$-norm for NRIC with reference shape $\bar{z}$, p. 3 |

**Neural Networks**

| | |
|---|---|
| $N_t$ | Layer sizes of a network, p. 4 |
| $T$ | Number of layers of a network, p. 4 |
| $\rho$ | Nonlinear activation function, p. 4 |
| $\zeta$ | Network parameters, p. 4 |
| $W_i^t$ | Matrix of learnable weights, p. 4 |
| $b^t$ | Learnable bias, p. 4 |
| $\mathcal{J}, \mathcal{J}_j$ | Loss functions used for training, p. 6 |
| $\psi_j^\zeta$ | Learnt parametrization of the factor manifolds, p. 5 |
| $\Psi^\zeta$ | Learnt map combining the outputs of the $\psi_j^\zeta$, p. 5 |

**Data Manifolds**

| | |
|---|---|
| $\mathcal{M}$ | Riemannian data manifold, p. 4 |
| $\mathcal{M}_i$ | Factors of a product manifold, p. 5 |
| $\Phi$ | Parametrization of a data manifold, p. 4 |

