# OpenReview forum: "Parametrizing Product Shape Manifolds by Composite Networks"
_ICLR.cc/2023/Conference — ICLR 2023 notable top 25%_

### Official Review · Reviewer_SjVS · 2022-10-25

**Confidence:** 3
**Correctness:** 4
**Technical Novelty And Significance:** 3
**Empirical Novelty And Significance:** 3
**Recommendation:** 8

**Clarity, Quality, Novelty And Reproducibility:**

See strengths and weaknesses. Very good clarity and novelty. Unsure about reproducibility especially for very generic shape manifolds.

**Strength And Weaknesses:**

Strengths

- I find the problem setup very original, novel, and interesting. The arguments that motivate the use of neural networks to model computationally expensive exponential maps are very refreshing
- Overall - the writing of this paper is fantastic despite some missing necessary details (see weaknesses) and occasional lack of background

Weaknesses, and Questions

- It is unclear what the role of the sparsity is in the process? Can it be said that the proposed approach cannot work for shape spaces having factors of variation that are not necessarily spatially localized? If so this seems restrictive. To ask a more direct question - does the proposed approach only work for some conceptually similar version of the freaky torus?
- I find the background on NRIC coordinates to be very short. To that aid, I think it is wise to provide a little more background on the time-discretization by Rumpf&Wirth (2015) for the generation of the exponential and logarithmic maps and hence to have at least some insight as to why that is a computationally expensive affair.
- On the practical side, it would be much more convincing to see interpolation/smoothly varying factors like in Fig 6 for more practical manifolds shown earlier.
- Figures 1,3 and 4 merit a better explanation. How does a more uniform density of points be indicative of a better model fit? And it is also important I think to also see the output of Sassen2020b for these figures (probably mention runtimes as well). In Table 1 what is the methodical difference between column 5 (Affine) and column 8 (Sassen 2020b) ?
- There is very little evaluation reported on the computational gains of the proposed method
- (suggestion) It would be interesting to explore some way to measure the generalization of the proposed approach, especially as a function of the size of training data. How does the quality of the fit change as I progressively increase the training samples from say 100 to 1000?

**Summary Of The Paper:**

This paper explores a learning-based approach for computing exponential maps for shape spaces modeled as Riemannian manifolds. Previous works have shown that statistical shape spaces represented as meshes can be modeled effectively using the NRIC  (Non-Rigid Invariant Coordinates)  system, allowing for operations like Principal Geodesic Analysis that encapsulate the dominant factors of variation in the corpus of shapes. However, the more meaningful tasks like shape interpolation require a computation of the exponential map which is arguably costly.

The authors propose a framework for computing the exponential map of specific shape manifolds - namely those that can be factored and written as product manifolds of much smaller dimensions (as required by conditions 0,1,2). The architecture is decomposed to look structurally identical to an affine sum of exponential maps of the submanifolds. By training the composite network to follow the idealistic exponential map in the training dataset, the authors demonstrate that there is merit in their approach, and show R2 confidence scores compared with a straightforward network and the direct axiomatic competitor (Sassen2020b). Overall the perception is that the network generates impressive models of shape variation at a (possibly?) much lower memory and compute cost.


**Summary Of The Review:**

Overall I find this to be a very interesting work, mainly for the novelty of the idea and clarity of discussion. However, the results seem to give an impression of “proof of concept” rather than convincing practical use cases. Put together I am weighing for a weak accept.

---

> ### Author Response · Authors · 2022-11-11
> **Author's Initial Response to Reviewer SjVS**
>
> Thank for the kind words on our paper and your helpful remarks. We agree that we can do a better job in explaining the background and making the practical advantages of our method more explicit.
> Concretely, we plan to complete the following tasks related to you comments in the next days:
>
> * The freaky torus is a simple example of scale separation (as the deformation of the global geometry of the torus is not sparse), whereas the SPGA is used to disentangle dominant modes of variability based on sparsity.
> We will study the decomposition into sparse modes and the corresponding localization in more detail in a new example section in the appendix.
> * We will briefly describe the relevant operators from time continuous and discrete Riemannian calculus in a new section of the appendix. This will also a include an overview table of used notation.
> * We will add, in the appendix, a short review on NRIC coordinates consisting of the vector of edge lengths and dihedral angles.
> * We will improve the figure's captions by explaining the finding of the respective figure more explicitly.
> * We will add a new section on additional experimental validation in the appendix. It will include a study on how the results depend on the amount of training data, a remark on computational gains in terms of runtime, and the above mentioned examples on the localization of factors.
> * We will add supplementary videos that show the quality of the exponential map extrapolation in animation applications.
>
> We hope that you agree this will further improve our paper. We will report back as soon as we have made modifications to the submission.

---

### Official Review · Reviewer_ADhs · 2022-10-25

**Confidence:** 3
**Correctness:** 3
**Technical Novelty And Significance:** 3
**Empirical Novelty And Significance:** 3
**Recommendation:** 8

**Clarity, Quality, Novelty And Reproducibility:**


The paper is mostly clear in the geometric tools that it uses. However, I feel that the overall goal can be better described. It can state the objective function upfront and use that to motivate the problem they are solving. The discussion about PGA, SPGA, etc, are useful but not necessary to understand the main goal.

**Strength And Weaknesses:**

Strengths

If the idea and approximation hold, then there is an efficient way to parameterize elements of a complex manifold.

Weakness:

There are a number of questions that arise from the paper. Perhaps it is my lack of understanding of the paper.

--I am not quite sure when the approximation is feasible and meaningful? Especially given how the training data is generated. I get the impression that the paper is trying to approximate:
Exp_z(v1 + v2 + …+ vn)
By the quantity
Psi(Exp_z(v1) \times Exp_z(v2) \times …. \times Exp_z(vn)).
Except that the latter factors are learnt from neural networks.
If this is correct, perhaps there is a simpler way to state the problem.

--- Can the authors provide an analytical nontrivial example where the factorization along principal directions is feasible?

-- I assume that in most interesting manifolds this factorization is not valid but perhaps one can make a good approximation. Can one always do that? When should we expect the approximation to be good? This brings us to the three assumptions.

-- The paper states several assumptions which seem plausible but not verified in the experimental setup. In the set up – “We assume that the manifold can be smoothly approximated by a product of much lower dimensional manifolds….” When does this work and not work?

-- For the shape manifolds studied here, do these assumptions hold? How can we verify them? For instance, the statement that “The different factor manifolds correspond to different spatial regions”. Is this forced or is it a fortuitous outcome of SPGA? Also, the local shape variations are claimed to be “independent” of global variations. Again, why?






**Summary Of The Paper:**

This paper seeks to approximate a given high-dimensional manifold with a product of some smaller, simpler manifolds. It learns this approximate mapping using neural network based optimization. The paper motivates this approach by discussing the concept of PGA (which is essentially Tangent PCA as stated here) and a sparse Tangent PCA. The idea is to approximate the range space of exponential map from the dominant tangent subspace by a composition of some smaller maps \psi_l. Both the smaller maps and their composition \Psi are learnt from the data using fully connected CNNs. The paper provides some experimental validation using shape space of surfaces. It generates training data using space PGA or TPCA and exponential maps of the individual components. It then compares the quality of approximation with some other methods.

**Summary Of The Review:**


To the extent I understand the presentation, the paper is trying to find efficient ways to explore complex manifolds. The approach is ambitious (learn the overall exponential map using a CNN). However, it is not so clear when the approach will be work and when it will not. It would have been useful to take some simpler intuitive examples where the approach succeeds and fails, and discuss those cases. Then, move on to more complicated shape manifolds. The validity assumptions need to be discussed further.

---

> ### Author Response · Authors · 2022-11-11
> **Author's Initial Response to Reviewer ADhs**
>
> Thank you for reading our paper carefully and providing us with helpful remarks to improve it.
> You are correct in your understanding of our goal and we will improve our paper to make this more apparent.
> Concretely, we plan to complete the following tasks related to you comments in the next days:
>
> * To better demonstrate the feasibility of the approximation of the exponential map via decomposition (based for instance on the SPGA) and composite networks, we are going to prepare some video sequences  showing the resulting animations and demonstrating the effectiveness of the decomposition.
> * A nontrivial, analytical example in very strict mathematical terms appears to be out of reach.
> However, we can demonstrate that for interesting, more complex geometries the SPGA generates a strictly decoupled support of modes. This leads to an exponential map, which can be composed of the exponential maps corresponding to the different modes.
> In addition, we will also show in a further example when the decomposition does not work due to bad mode separation.
> * Indeed, the postulation of several assumptions and the experimental validation could be improved.
> We are going to add further examples in a new section in the appendix to justify these assumptions, in particular the
> proper decomposition of the exponential map.
> * Furthermore, we will better describe the overall goal. This will include a better description of the underlying continuous and discrete Riemannian calculus in a separate section in the appendix and a more precise description of the experimental findings.
> * We will add a table with used notation in the appendix to improve the readability of the paper.
>
> We hope that you agree this will strengthen our paper. We will report back as soon as we have made modifications to the submission.

---

### Official Review · Reviewer_VcAe · 2022-10-25

**Confidence:** 3
**Correctness:** 3
**Technical Novelty And Significance:** 2
**Empirical Novelty And Significance:** 2
**Recommendation:** 5

**Clarity, Quality, Novelty And Reproducibility:**

The clarity of this paper needs further improvement. Because of missing details, the soundness of the proposed method is doubtful and the reproducibility of this paper is low. The originality of the work is hard to evaluate due to the missing appropriate discussion on related work.

**Strength And Weaknesses:**

Strength:

+) This paper targets an interesting and challenging problem and the proposed method seems to make sense.

+) The qualitative and quantitative results demonstrate the effectiveness of the proposed method.

Weaknesses:

-) Missing a good literature review.  The related work should provide a summary of existing works that tackle the same or similar problem; while the related work in this paper looks like a background. A missing discussion on existing methods makes it unclear where this paper stands in the literature.

-) The equations in this paper are not clearly explained, which brings difficulty in understanding the details of this paper.  For example, in Eq. 1, what is \bar{z}? Is it the mean of a set of shapes? If so, how to calculate it? The same question for the defintion of R2 score. Constructing an atlas for shapes is also non-trivial. In Eq. 3, what is l_e^b, l_e^a, etc.? In Eq.5, what is Z_e^0? It is unclear what is the input and output of the neural networks. The readers have to guess, perhaps explicitly pointing this out would make the paper more understandable and readable.

-) In the loss function J_l(\theta), what is exp operator? Is "a" a vector? then what is S^l? a set? but it says "a random sample"? In J(\theta), the \sum_\omega_i(a^i) is the sum of tangent vector. Should it be the composition of several exp operators? When a point moves on manifolds, it goes step by step and each is a small step to make sure it stays on manifolds. How to understand the definition of J(\theta)?

**Summary Of The Paper:**

This paper works on learning the exponential map of a shape space using deep neural networks.  The proposed method adopts a product structure to approximate shape manifolds using a sum of multiple low-dimensional submanifolds. The method is evaluated on synthetic data and manifolds extracted from data using sparse principal geodesic analysis.

**Summary Of The Review:**

The experimental results look good; however, the current shape of this paper is not ready for publishing. More efforts are desired to make this paper easy to understand and use by other researchers.

---

> ### Author Response · Authors · 2022-11-11
> **Author's Initial Response to Reviewer VcAe**
>
> Thank you for your careful reading of the manuscript and the helpful comments to improve the description of our method and the presentation of the results.
> Indeed, we build on a wealth of previous work, which entails some intricate notation. Your comments will be helpful to make this more accessible for future readers.
> Concretely, we plan to complete the following tasks related to you comments in the next days:
> * We are going to improve the literature review and emphasize what is general methodological background literature and what are to the best of our knowledge the few papers which share a similar ansatz or a similar goal. We hope that this also clarifies the original contributions of our manuscript.
> * To improve the readability of the manuscript, we are going to add a table with notation and we carefully check once more if all terms appearing in formulas are previously defined. We expect that this also better explains the chosen loss functional $J(\theta)$.
> * Our manuscript is based on the continuous and discrete Riemannian calculus. We are going to add a section in the appendix, which gives a brief overview of the basic geometric tools and their time discretization. In particular, we will explain the exponential map `exp' and its discrete counterpart, which are at the core of our method, in more detail.
> * We are going to reshape the discussion of the results and explain the finding more explicitly.
>
> We hope that we will address your concerns this way. We will report back as soon as we have made modifications to the submission.

---

### Author Response · Authors · 2022-11-17
**Revision uploaded**

Dear reviewers,

We have uploaded the revised version of our manuscript. First, thank you again for your thoughtful remarks. They helped us improve our paper and make it better accessible for future readers. As announced, we made the following major changes in the revision:
 * We added new experiments in the appendix demonstrating the application of our method to animation problems and explaining the computational advantages arising from it. This is also shown in the supplementary video.
 * We added new illustrations and explanations justifying the postulated assumptions for the considered shape data manifolds.
 * We added more detailed explanations of Riemannian operators and their discretization in the appendix.
 * Also in the appendix, we added a more detailed overview of NRIC.
 * In the main body, we made several changes to better (and earlier) explain our goals, position ourselves better relative to existing literature, and describe our results/figure more explicitly.
 * We added a table of used notation in the appendix and fixed some notational problems that might confuse readers.

For your convenience, we colored modified and added parts of the paper in blue. Furthermore, we fixed some typos here and there, which are not necessarily highlighted.
We hope you agree that the revised manuscript marks a considerable improvement over the original submission.

---

### Decision · Program_Chairs · 2023-01-20

**Decision:**

Accept: notable-top-25%

**Justification For Why Not Higher Score:**

This paper could provide some really novel and refreshing input to the learning on Riemannian manifolds community -- but this is a small community and I suppose I would expect even more of a paper that makes it to an oral presentation.

**Justification For Why Not Lower Score:**

The approaches provided in the paper could really make an impact in the learning on Riemannian manifolds community.
I am voting "this decision can be bumped down", but ONLY to poster -- this is a clear accept.

**Metareview: Summary, Strengths And Weaknesses:**

Summary:
This paper aims to approximate Riemannian manifolds with neural networks as products of lower-dimensiona, simpler manifolds.

Strengths:
- The paper is very well written
- The paper has an interesting idea and experiments supporting its utility
- For those cases where the approximation works, it provides a very efficient approximation

Weaknesses:
- The related work could be stronger
- There are some questions regarding the paper's clarity, in particular some of the equations
- The computaitonal gains of the method could be better explored experimentally

**Note From Pc:**

if the above contains the word "oral" or "spotlight" please see: "oral" presentation means -> notable-top-5% and "spotlight" means -> notable-top-25%. As stated in our emails, we are disassociating presentation type from AC recommendations